# Activated Eosinophils Predict Longer Progression-Free Survival under Immune Checkpoint Inhibition in Melanoma

**DOI:** 10.3390/cancers14225676

**Published:** 2022-11-18

**Authors:** Nadine L. Ammann, Yasmin F. Schwietzer, Christian Mess, Julia-Christina Stadler, Glenn Geidel, Julian Kött, Klaus Pantel, Stefan W. Schneider, Jochen Utikal, Alexander T. Bauer, Christoffer Gebhardt

**Affiliations:** 1Department of Dermatology and Venereology, University Medical Center Hamburg-Eppendorf (UKE), 20246 Hamburg, Germany; 2Fleur Hiege Center for Skin Cancer Research, University Medical Center Hamburg-Eppendorf (UKE), 20246 Hamburg, Germany; 3Department of Tumor Biology, University Medical Center Hamburg-Eppendorf (UKE), 20246 Hamburg, Germany; 4Skin Cancer Unit, German Cancer Research Center (DKFZ), 69121 Heidelberg, Germany; 5Department of Dermatology, Venereology and Allergology, University Medical Center Mannheim, Ruprecht-Karl University of Heidelberg, 68167 Mannheim, Germany; 6DKFZ Hector Cancer Institute at the University Medical Center Mannheim, 68167 Mannheim, Germany

**Keywords:** melanoma, immune checkpoint inhibition, eosinophils, ECP, EPX, T-cells, inflammation, tumor microenvironment, biomarker

## Abstract

**Simple Summary:**

Immune checkpoint inhibitors, which stimulate the patient’s own T-cells to attack tumor cells, have revolutionized the treatment of metastatic melanoma. However, not all melanoma patients respond to therapy possibly due to a lack of T-cells present in or entering tumor tissue. It is presumed that eosinophils could aid T-cell-mediated immune response against tumor cells. In order to describe the local association of eosinophils and T-cells within the tumor microenvironment we investigated specific markers for cell type and activation status using immunofluorescence. Additionally, blood measurements were performed to determine the effects of eosinophil count and their activation status on the efficacy of immune checkpoint inhibition. There was a strong correlation between activated eosinophils and T-cells in melanoma. Furthermore, patients with high blood levels of activated eosinophils showed a delayed tumor progression. In the future, eosinophils may serve as prognostic biomarkers as well as novel therapeutic targets in melanoma.

**Abstract:**

Immune checkpoint inhibition (ICI) has yielded remarkable results in prolonging survival of metastatic melanoma patients but only a subset of individuals treated respond to therapy. Success of ICI treatment appears to depend on the number of tumor-infiltrating effector T-cells, which are known to be influenced by activated eosinophils. To verify the co-occurrence of activated eosinophils and T-cells in melanoma, immunofluorescence was performed in 285 primary or metastatic tumor tissue specimens from 118 patients. Moreover, eosinophil counts and activity markers such as eosinophil cationic protein (ECP) and eosinophil peroxidase (EPX) were measured in the serum before therapy start and before the 4th infusion of ICI in 45 metastatic unresected melanoma patients. We observed a positive correlation between increased tumor-infiltrating eosinophils and T-cells associated with delayed melanoma progression. High baseline levels of eosinophil count, serum ECP and EPX were linked to prolonged progression-free survival in metastatic melanoma. Our data provide first indications that activated eosinophils are related to the T-cell-inflamed tumor microenvironment and could be considered as potential future prognostic biomarkers in melanoma.

## 1. Introduction

The discovery of immune checkpoints on T-cells, such as cytotoxic T-lymphocyte-associated antigen 4 (CTLA-4) or programmed cell death protein 1 (PD-1), and using them as therapeutic targets marked a breakthrough in melanoma treatment [1]. The implementation of immune checkpoint inhibition (ICI) via anti-CTLA-4 antibodies (ipilimumab) and anti-PD-1 antibodies (pembrolizumab or nivolumab) resulted in T-cell reactivation and tumor control [2,3]. In particular, anti-PD-1 monotherapy and the combination therapy of ipilimumab and nivolumab were able to prolong both long-term progression-free survival (PFS) and overall survival (OS) in advanced melanoma patients [4,5]. Such outcomes had previously been inconceivable and establish ICI as the gold standard in the first-line treatment of metastatic melanoma.

Unfortunately, not every patient with stage III and IV melanoma (according to the American Joint Committee of Cancer (AJCC) classification) who receives ICI benefits from it. In fact, only 58% of melanoma patients treated with ipilimumab plus nivolumab showed a clinical response and 55% had immune-related adverse events (irAEs) of grade 3 or 4 [6]. The cause of these irAEs is thought to be autoreactive T-cells, which can potentially attack any organ and lead to severe inflammation [7,8]. The most common inflammatory irAEs during combined anti-PD-1 and anti-CTLA therapy were diarrhoea and colitis, which even led to discontinuation of ICI in up to 38% of cases [6]. However, a correlation between the occurrence of irAEs and higher efficacy of anti-PD-1 treatment was found, suggesting a link between inflammatory settings and the effectiveness of ICI [9,10].

A tumor’s particular composition of proinflammatory cytokines and immune cells determines how immunoactive (“hot”) the tumor is. Hot tumors are characterized by a high infiltration of T-cells and an inflammatory tumor microenvironment (TME). They are associated with better response to ICI than tumors without T-cell infiltrate (“cold” tumors) [11,12,13]. On the one hand, understanding which soluble and cellular components characterize a hot tumor could help to predict which patient may respond to ICI. On the other hand, future research could aim at converting cold tumors into hot tumors in order to improve treatment outcomes [12,14].

In search of factors that determine a “hot” T-cell-inflamed TME in melanoma and are crucial for the response to ICI, several T-cell-specific chemokines, such as C-C motif chemokine ligand (CCL) 5, C-X-C motif chemokine ligand (CXCL) 9 or CXCL10, were discovered [15,16,17]. Indeed, these chemokines can be secreted by eosinophils and subsequently lead to a recruitment of cluster of differentiation (CD) 8+ effector T-cells into the tumor, which was recently shown in murine melanoma experiments [18,19]. There are indications that eosinophil activation may be essential to promote the anti-melanoma T-cell response [19]. We hypothesize that activated eosinophils affect disease progression and ICI outcome by stimulating T-cell responses against tumor cells.

Whether activated eosinophils generate an anti-tumoral T-cell response in melanoma and hence influence tumor progression was investigated in the first part of this study in 118 melanoma patients not receiving ICI. Immunofluorescence co-staining of eosinophils and CD8+ effector T-cells in tissue sections of 77 nevi (control), 108 primary melanomas and 177 associated metastases allowed for the analysis of both infiltrating cell populations. For this purpose, we used the established eosinophil marker sialic acid-binding Ig-like lectin 8 (Siglec-8) and the two common activity markers eosinophil cationic protein (ECP) and eosinophil peroxidase (EPX), which were then also tested for their prognostic value [20,21,22,23].

To examine the influence of eosinophils and their activity markers on the effectiveness of ICI, we measured eosinophil, ECP and EPX concentrations in the peripheral blood as the second part of the study. Serum samples of 45 unresected stage III and IV melanoma patients were analyzed before administration (baseline) and during ICI with anti-CTLA-4 and anti-PD-1 antibodies.

The aim of this work was to clarify the impact of eosinophils on tumor progression and the efficacy of ICI in melanoma. We investigated whether the presence of eosinophils can serve as a prognostic marker and whether this could predict the clinical outcome of ICI.

## 2. Materials and Methods

### 2.1. Patient Collective and Clinical Data

This study consists of two parts: (1) local expression analyses of tested biomarkers using immunofluorescence co-staining of tissue microarrays (TMA) and (2) systemic investigations of the respective biomarker concentrations in the serum of metastatic melanoma patients based on enzyme-linked immunosorbent assays (ELISA).

(1) The retrospective tissue analyses of nine TMAs included a total of 362 tissue samples from 118 cutaneous melanoma patients of all stages according to the 8th edition of the AJCC melanoma staging system [24]. Among all nine TMAs, 77 melanocytic nevi, 108 primary melanomas and 177 associated metastases were represented (for tissue characteristics see Table 1).

Inclusion criteria comprised all histological cutaneous melanoma types. Exclusion criteria were mucosal and uveal melanoma, minority, presence of autoimmune disease, acute viral infections such as human immunodeficiency virus (HIV), hepatitis B or C, pregnancy and concomitant systemic melanoma therapy.

Melanoma patients were treated at the Department of Dermatology, Venereology and Allergology at the University Hospital Mannheim and did not receive any therapy prior tissue biopsy. With the approval of the local ethics committee of the University Medical Center Mannheim (ethics votes 2010-318N-MA and 2014-835R-MA), each patient gave written consent for the tissue biopsies and patient-related data to be collected for research purposes over a seven-year period.

(2) For ELISA measurements, peripheral blood samples from 45 metastatic unresected melanoma patients (AJCC 2017 stage III/IV) receiving ICI at the University Skin Cancer Centre Hamburg (Department of Dermatology and Venereology, UKE) were utilized. All patients provided written informed consent for the usage of any clinical and laboratory data. The Ethics Committee of the Hamburg Medical Association approved this study concept (ethics vote PV5392). In addition, pooled serum from eight healthy volunteers served as controls, matching the average gender and age distribution.

At the Skin Cancer Center Hamburg, patients received either pembrolizumab (10 mg per kg body weight) every 3 weeks, nivolumab (240 mg) every two weeks or a combination of nivolumab (1 mg per kg body weight) and ipilimumab (3 mg per kg body weight) every 3 weeks. Only patients with serum samples available at both measurement time points, before the first ICI administration and before the fourth infusion cycle of ICI, were recruited to the study. The inclusion criteria further included being of legal age, no melanoma-specific therapy within 28 days prior initiation of ICI, and any histological melanoma types, inclusive mucosal and uveal melanoma. Exclusion criteria were the presence of acute viral infections such as HIV, hepatitis B or C, autoimmune disease and pregnancy.

After treatment initiation, response to therapy was monitored every 12 weeks by contrast-enhanced computed tomography (CT) of the whole body, magnetic resonance imaging (MRI) or positron emission tomography (PET-CT). According to the current Response Evaluation Criteria in Solid Tumors (RECIST 1.1), clinical responses were defined radiologically as complete response (CR), partial response (PR), stable disease (SD) or progressive disease (PD) [25,26]. The best overall response determined the classification of patients into responders (CR, PR) and non-responders (SD, PD).

### 2.2. Immunofluorescence Co-Staining of Eosinophils and Effector T-Cells in Tissue Microarrays

Formalin-fixed paraffin sections of nine TMAs were used for immunofluorescence staining, representing a total of 77 melanocytic nevi, 108 primary tumors and 177 associated metastases. Initially, the paraffin sections (1-µm-thick) underwent rehydration in a descending ethanol series and the antigen epitopes were subsequently unmasked in a citrate buffer (0.01 M and pH = 6) in the microwave at 360 watts. A 0.01% trypsin solution was applied for the final enzymatic unmasking and a protein block (X0909, Dako, Santa Clara, United States) for saturation of the non-specific antibody binding capacity. The following primary antibodies were added and incubated overnight: anti-Siglec-8-antibody (ab198690, Abcam, Cambridge, UK), anti-ECP-antibody (ab207429, Abcam, Cambridge, UK), anti-EPX-antibody (ab238506, Abcam, Cambridge, UK) and anti-CD8-antibody (IR623, Dako, Santa Clara, United States). Alexa Fluor 488 (A11034, Invitrogen, Waltham, United States) was then used as a secondary antibody to visualize the stained eosinophils, and Alexa Fluor 555 (A21422, Invitrogen, Waltham, United States) for the stained effector T-cells. Nuclear staining was conducted in blue with a 0.02% 4′,6-Diamidin-2-phenylindol (DAPI) solution (10236276001, Roche, Mannheim, Germany). To prevent false positive and negative results, a negative control without adding the first antibodies and a positive control were performed.

### 2.3. Microscopic Analysis of Fluorescent Stained Tissue Samples

In the context of immunofluorescence microscopy, each stained tissue sample was analyzed with a laser microscope (Zeiss Axio Observer.Z1) and its high-resolution 40× oil immersion objective. The nine TMAs were evaluated manually in a blinded fashion by two dermatohistopathologists. Therefore, objective criteria for counting were established in advance, such as the presence of the typical nuclear form, the signal reference to the nucleus, the signal strength standing out from the background and the corresponding signal pattern. Counted cells were divided by the measured area to achieve a standardized comparable cell count.

### 2.4. Determination of ECP and EPX Serum Levels

Serum was collected the day before the first administration of ICI (baseline) and before the fourth infusion cycle (4th cycle, C4) using gel-coated serum tubes (Sarstedt, Nümbrecht, Germany). After unrestrained centrifugation at 1000× *g* for 10 min, the serum samples were stored at −80 °C. Serum levels of ECP and EPX were measured in triplicates according to the standards of commercially available sandwich ELISA kits (ECP ELISA kit: 7618E, MEDICAL & BIOLOGICAL LABORATORIES, Nagoya, Japan; EPX ELISA kit: EH2381, Whuan Fine Biotech, Whuan, China). Eosinophil counts were determined during routine clinical laboratory examinations.

### 2.5. Statistical Analysis

Statistical calculations were performed with R (version 4.1.1) and GraphPad PRISM (version 9). Calculated results with a *p*-value of less than or equal to 0.05 were defined as statistically significant.

For group comparison of the quantified cells or measured serum concentrations, one-way analysis of variance on ranks was performed, whereby *p*-values were calculated according to the Kruskal–Wallis test and the Wilcoxon-Mann–Whitney test was used for post hoc comparisons. All correlations were evaluated using Kendall’s rank correlation coefficient (Kendall’s tau). The univariate analyses of survival times were presented in form of Kaplan–Meier curves. For each survival analysis, a cut-off was calculated by optimizing the *p*-value of the log-rank statistics. Endpoints were progression-free survival (PFS), i.e., the time from study entry to disease progression, and overall survival (OS), which is defined by the time period between study entry and patient death. Patients who did not die or did not experience tumor progression were censored at the last assessment time point.

## 3. Results

### 3.1. TMA Expression Analysis—Patient Characteristics

A total of 108 primaries, 77 corresponding melanocytic nevi, and 177 associated metastases from 118 melanoma patients who did not receive systemic therapy were recruited for the local expression analyses. In this cohort, 76 males (64.4%) and 42 females (35.6%) were included and the median age was 65.8 years (standard deviation of 13.5 years). According to the 8th AJCC classification, 17 patients (14.4%) had stage I, 61 patients (51.7%) had stage II, 32 patients (27.1%) had stage III and 8 patients (6.8%) had stage IV melanoma.

### 3.2. Quantification of Tumor-Infiltrating Eosinophils

It is still not completely elucidated which role eosinophils play in the progression of melanoma and whether they have pro- or anti-tumoral effects. In order to characterize their impact on the course of the disease and to assess whether their counts have prognostic implications, their expression was quantified in three different tissues: melanocytic nevi (as controls), primary tumors and metastases of melanoma patients (Appendix A). Eosinophils were labelled with an anti-Siglec-8 antibody in the immunofluorescence stainings, while their activity state could be detected using antibodies against degranulated ECP and EPX.

We observed a significantly higher number of Siglec-8+ eosinophils (*p* < 0.0001, Appendix A), as well as activated ECP+ (*p* < 0.0001, Figure 1d) and EPX+ (*p* < 0.0001, Appendix A) eosinophils in the primaries and metastases, compared to the nevi. In contrast, the expression of the three eosinophil markers did not differ significantly between primaries and metastases (Figure 1d, Appendix A). As an example, there was five times less ECP expressed in nevi (Figure 1a) compared to primaries (Figure 1b) and four and a half times less than in metastases (Figure 1c). Expression of ECP was 8 % higher in primary tumors than in metastases and hence barely differed (Figure 1d).

In summary, an increased infiltration and activation of eosinophils was seen in melanoma tissues (Figure 1b,c, Appendix A). This does not differ significantly between the different stages of progression, such as between primary tumors and metastases in advanced melanoma.

### 3.3. Quantification of Tumor-Infiltrating Effector T-Cells

To investigate whether the infiltration of eosinophils was related to the infiltration of effector T-cells, nevi (Figure 2a), primaries (Figure 2b) and metastases (Figure 2c) were additionally stained with antibodies against CD8. This allowed an assessment of the amount of T-cells in the same tissue sections, using the mean of the counted T-cells in all three co-stainings (Siglec-8 + CD8, ECP + CD8, EPX + CD8). A significant six-fold reduction in CD8 expression was seen in nevi compared to primaries and a eight-and-a-half-fold reduction in expression compared to metastases. The CD8+ T-cells were 26% more abundant in the metastases than primaries and thus there was no significant difference between primaries and metastases (Figure 2d).

### 3.4. Correlation of Tumor-Infiltrating Eosinophils and Effector T-Cells in Melanoma

Since infiltration of Siglec-8, ECP and EPX expressing eosinophils, as well as CD8+ effector T-cells could previously be observed in the tissue samples of melanoma patients, their correlation was further investigated. In all tissue samples, a positive correlation existed between the amount of infiltrating eosinophils and effector T-cells. The expression of all eosinophil markers (Siglec-8, ECP, EPX) correlated with the level of the expressed T-cell marker (CD8), as shown in Table 2.

As an illustration, an associated metastasis of melanoma is depicted below, which shows a high infiltration of ECP expressing eosinophils in combination with CD8 positive effector T-cells (Figure 3). Further example images of the coherent expression of the other eosinophil markers (Siglec-8, EPX) and T-cell marker are shown in Appendix A.

### 3.5. Association between the Amount of Tumor-Infiltrated Activated Eosinophils as Well as Effector T-Cells and the Survival of Melanoma Patients

Survival analyses were performed to examine how eosinophil and T-cell infiltration in the primaries affects melanoma progression. Higher counts of ECP+ eosinophils (cut-off 81.26 cells per mm^2^, Figure 4a) and CD8+ effector T-cells (cut-off 61.89 cells/mm^2^, Figure 4b) were related to prolonged PFS in primary melanoma. In contrast, the level of expressed Siglec-8 (Appendix A) and EPX (Appendix A) in the primary tumors had no influence on PFS.

Similar analyses were conducted on the corresponding nevi (Appendix A) and associated metastases (Appendix A) of the melanoma patients, where an opposite effect was found. Higher expressions of Siglec-8, ECP, EPX and CD8 were associated with accelerated progression here (Appendix A).

Taken together, the extent of T-cell infiltrates correlates positively with infiltrating eosinophils in melanoma. A high proportion of infiltrated activated (ECP+) eosinophils as well as CD8+ effector T-cells in primary melanomas had a favourable impact on the prognosis of melanoma patients. The way in which eosinophil count and activation status might affect ICI treatment efficacy was then further investigated in the blood-based analysis during ICI.

### 3.6. Blood-Based Analysis during ICI—Patient Characteristics

For systemic analyses, the peripheral blood of 45 metastatic unresected melanoma patients treated with ICI was analyzed (Table 3). Among them were 32 males (71.1%) and 13 females (28.9%) with a median age of 60.9 (standard deviation of 18.2 years). Based on the current AJCC classification (8th edition), 4 patients were classified as unresectable stage III melanoma and 41 as unresectable stage IV melanoma. Among the cohort were 4 metastatic uveal melanoma and no mucosal melanoma. ICI was administered to all patients; 6 (13.3%) received monotherapy with nivolumab and 11 (24.5%) with pembrolizumab, while 28 (62.2%) received ipilimumab combined with nivolumab. Before the start of ICI, 16 patients received another systemic therapy. 26 patients experienced irAEs, of which 16 (35.6%) were grade I or II and 10 (22.2%) were grade III or IV. Among them, two received a steroid shot together with infliximab, one received two steroid shots and one received carbimazole to alleviate the side effects. Treatment response was assessed by the RECIST 1.1 criteria, with 13 (28.9%) melanoma patients achieving a CR and 16 (35.6%) a PR. These patients were considered as responders, while 5 (11.1%) melanoma patients with a SD and the 11 (24.4%) patients with PD were defined as non-responders.

### 3.7. High Absolute Eosinophil Count and Elevated ECP Serum Levels Prior to ICI Initiation Are Related to Delayed Progression

To determine whether increased numbers of activated ECP-expressing eosinophils in the primary melanomas translate into increased numbers of degranulated ECP in the serum and which prognostic impact this has on ICI, a second cohort was examined (Table 3). For this purpose, we measured the absolute eosinophil count (AEC) in the peripheral blood as well as the serum level of ECP at two different time points–before ICI (baseline) and before the fourth infusion cycle of ICI (C4)–and performed univariante survival analyses (Figure 5).

Patients with high baseline AEC (>0.13 × 10^9^ cells/L, Figure 5a) and high ECP (>37.58 ng/mL, Figure 5b) serum levels showed significantly prolonged PFS compared to those with low baseline AEC and ECP levels. In contrast, a correlation between increased C4 AEC (>0.03 × 10^9^ cells/L, Figure 5c) and ECP (>9.47 ng/mL, Figure 5d) levels and earlier disease progression was found during therapy, but was not significant.

The same concentration measurements were also carried out with EPX, the second eosiophil activity marker. Melanoma patients with an elevated (>1.07 ng/mL) baseline EPX serum level also showed a significant delay in progression (Appendix A), while the level of EPX serum at C4 had no effect on PFS (Appendix A).

In addition, survival analyses for OS of the 45 metastatic melanoma patients were evaluated using peripheral-blood AEC (Appendix A), ECP (Appendix A) and EPX (Appendix A) serum levels at BE and C4, as shown in Appendix A. A significantly prolonged OS was observed for melanoma patients with reduced C4 AEC levels (Appendix A) as well as elevated baseline and C4 EPX serum levels (Appendix A).

To summarize, in metastatic melanoma, a high number of eosinophils and strong eosinophil activation in the blood is associated with slower disease progression in patients receiving ICI.

### 3.8. Constant to Decreasing AEC and ECP Levels between Baseline and the Fourth Infusion Cycle of ICI Are Associated with Later Progression of Metastatic Melanoma

To explore whether the trend in AEC and ECP serum levels during ICI is relevant to treatment outcome, progression analysis was performed. Therefore, the absolute change in AEC and ECP values between C4 and BE was calculated and univariate survival analyses for the PFS of the 45 patients with metastatic melanoma were performed based on the AEC or ECP differences.

Melanoma patients with decreasing (cut-off −0.06 × 10^9^ cells/L) concentrations of AEC between baseline and the fourth infusion cycle of ICI achieved a prolonged PFS compared to patients with increasing peripheral-blood AEC (Figure 6a). Likewise, constant to decreasing (cut-off 3.46 ng/mL) ECP serum levels during ICI were associated with delayed progression in stage III and IV melanoma patients (Figure 6b). The same was done for EPX, but no significant result was obtained (Appendix A).

Similar survival analyses were conducted for the OS of the metastatic melanoma patients, with decreasing (>0.08 × 10^9^ cells/L) concentrations of AEC during ICI, showing an association with prolonged OS (Figure 6c). Instead, the kinetics of ECP (Figure 6d) and EPX (Appendix A) levels between baseline and C4 had no significant effect on OS in this patient cohort.

### 3.9. Responders and Non-Responders Do Not Differ in Peripheral Blood AEC, as Well as ECP and EPX Serum Levels

Next, the predictive value of AEC, ECP and EPX was evaluated. We divided the 45 patients with metastatic melanoma into responders (including CR and PR) and non-responders (including SD and PD), and compared baseline AEC, ECP and EPX within both groups. Analysis of variance (one-way ANOVA) revealed no significant difference neither in baseline peripheral-blood AEC (Appendix A) nor ECP (Appendix A) and EPX (Appendix A) serum levels between responders and non-responders.

Corresponding ANOVA analyses were conducted for the C4 time point. During ICI, there were likewise no differences in C4 levels of AEC (Appendix A), ECP (Appendix A) and EPX (Appendix A) between responders and non-responders.

## 4. Discussion

Melanoma is considered one of the most immunogenic malignancies, which highlights the role of the immune system in tumor control [27,28]. As T-cells represent the effector cells of the immunological tumor defence, their infiltration is crucial for achieving an anti-tumoral response. It is hence of immense prognostic importance [29,30,31,32]. Detailed knowledge of the TME in individual melanoma patients and characterization of cell populations favouring an anti-tumor immune response are still lacking. Recently, it has been reported that eosinophils can facilitate the recruitment and infiltration of T-cells into mice bearing melanoma [18,19].

In the first part of this study, the co-occurrence of eosinophils and T-cells as well as their prognostic value were investigated using co-immunofluorescence staining in human melanoma sections. We observed a strong infiltration of eosinophils in the primary melanomas and associated metastases (Figure 1, Appendix A), which is consistent with studies in various other solid tumor entities [33]. The expression of eosinophil surface marker protein Siglec-8 correlated positively with CD8 of the effector T-cells (Table 2, Appendix A). Lucarini et al. had previously shown in a mouse model of melanoma that anti-Siglec-F antibody injection was associated with reduced tumoral recruitment of CD8+ effector T-cells, indicating reciprocal promotion of tumor infiltration [18]. The literature suggests that tumor necrosis factor (TNF)- and interferon (INF)γ-activated eosinophils secrete T-cell attracting cytokines such as CCL5, CXCL9 or CXCL10, inducing CD8+ T-cell migration into the TME and resulting in anti-tumoral T-cell response [19,34,35,36]. Accordingly, the activity status of the eosinophils seems particularly relevant for the crosstalk with T-cells. In order to determine whether eosinophils in human melanoma sections were also activated, we co-stained the eosinophil activity markers ECP and EPX with CD8. Again, a positive correlation was found between the infiltrated activated ECP as well as EPX expressing eosinophils and CD8 expressing T-cells (Table 2, Figure 3 and Appendix A). Thus, our study demonstrates a positive association between the infiltration of activated eosinophils and CD8+ T-cells in human melanoma tissue and is in line with a previous report in a small patient cohort [37]. Furthermore, none of the melanoma patients we recruited received systematic therapy before sampling, which renders a therapy-associated effect impossible.

Likewise, with regard to the prognostic potential of the infiltrated eosinophils, their activity state proved to be decisive. Melanoma patients with a high number of ECP expressing eosinophils and CD8+ T-cells in primary melanomas showed prolonged PFS (Figure 4). This observation could be explained by the anti-tumorigenic activity of eosinophils, which has been noted in numerous mice melanoma experiments [18,19,38,39]. Eosinophils mediate anti-tumor responses via direct and indirect mechanisms, e.g., by secreting tumor cytotoxic proteins, such as major basic protein (MBP), ECP, eosinophil-derived neurotoxin (EDN) and granzymes, as well as by limiting tumor cell migration through secretion of interleukin (IL)-12 and IL-10. Additionally, eosinophils are capable of remodelling the TME by expressing natural killer (NK)-cell-associated activation receptors, such as 2B4, NKG2D and LY49, and promoting anti-tumor immunity through release of IFNγ. Besides, they support the polarization of macrophages towards an anti-tumorigenic phenotype and anti-tumor immunity by normalizing the vasculature [33,34].

However, increased amounts of eosinophils and T-cells in metastases of melanoma patients were associated with earlier progression, suggesting an ambivalent role of eosinophils in tumor progression (Appendix A). This opposing trend of eosinophils in primary tumors versus in metastases of melanoma patients need to be re-evaluated in a larger cohort, and at the same time the impact on OS should be considered. Unfortunately, we could not perform survival analyses regarding the OS of the melanoma patients, as a too small proportion of patients died during the observation period. Eosinophils are regarded as heterogeneous immune cells that can polarize in different ways and thus act either pro- or anti-tumoral [22,40]. Therefore, eosinophils could be polarized differently depending on the time phase of tumor growth. More detailed research would be needed to identify the molecular structures in the TME that promote each type of polarization.

The favourable effect of ECP expressing eosinophils in melanoma primaries on prognosis observed in this study could be explained in two ways: in vitro, ECP has been shown to induce tumor lysis through its cytotoxic capacity, which may clarify the delayed progression in primary melanomas with high ECP expression, which we demonstrated [41,42,43]. On the other hand, the activated eosinophils could also have indirectly contributed to a sufficient T-cell response against the melanoma via increased T-cell recruitment, as previously demonstrated in melanoma-bearing mice [18,19]. Our analyses provide a first indication that co-staining of ECP and CD8 in primary melanomas could be used as a prognostic tool to better predict individual melanoma progression.

As melanoma is a rapidly growing tumor entity with a strong tendency to metastasize, such prognostic biomarkers are tremendously important [44]. In this context, biomarkers that can be measured in routine clinical laboratory tests are particularly useful for clinical practice. Consequently, in the second part of this study, we analyzed blood samples from metastatic melanoma patients to determine whether eosinophils and their activity markers ECP and EPX could act as prognostic biomarkers. Assuming that eosinophils contribute to a functioning anti-melanoma T-cell response, they might also enhance the efficiency of ICI, which led to the selection of a second cohort of melanoma patients receiving ICI (Table 3).

Elevated baseline peripheral-blood AEC was found to be associated with improved PFS in 45 unresected stage III and IV melanoma patients (Figure 5a). The baseline level of AEC, however, had no influence on the OS in our cohort (Appendix A), which contrasts with the study by Martens et al., where a correlation between OS of the 209 stage IV melanoma patients and baseline AEC was confirmed [45]. Analyses of two further large collectives of stage III and IV melanoma patients revealed that relative eosinophil counts correlate with OS [46,47]. As we wanted to test the prognostic effect of eosinophils within a cohort receiving all approved ICI therapy regimens, the discrepancy could be attributed to different patient cohorts and especially to differing follow-up periods. Our results would need to be further monitored over the current follow-up period of approximately 2.5 years. The patient population should be expanded to re-examine the impact of AEC on OS. Concerning a long follow-up period of 12 years, another study of 172 stage IV melanoma patients indicated that regardless of the ICI treatment and timing, any increase in eosinophils resulted in extended survival times [48]. This finding differs from our measurements during ICI, where an increase in AEC between baseline and the fourth infusion cycle was associated with shorter PFS and OS (Figure 6a,c). We are the first research group to observe that a decrease in peripheral-blood eosinophils during ICI led to later disease progression (Figure 6a). All other studies reported a prognostic advantage for melanoma patients with an increase in eosinophil count during ICI treatment [37,49,50]. Indeed, we demonstrated the association of an early increase in AEC with a better outcome after treatment with ipilimumab in previous research analyzing the peripheral blood of 59 stage IV melanoma patients [49]. In addition, higher AEC was measurable in responders, whereas in our analyses the amount of AEC had no influence on treatment response. The inclusion of other immune checkpoint inhibitors, different measurement time points and a varied distribution of responders and non-responders might be responsible for the fact that a predictive value of baseline AEC could not be verified here. Our results support the prognostic value of eosinophils but verification in a larger cohort at all infusion time points of ICI is needed.

To investigate whether activated serum eosinophils have a different effect and their degranulated ECP and EPX can be used as biomarkers, ELISA experiments were per-formed. Higher baseline serum levels of ECP were related to extended PFS in metastatic melanoma patients (Figure 5b). Reduced serum levels of ECP, on the other hand, tended to be associated with a longer OS (Appendix A). So far, only one other study has analyzed the prognostic effect of ECP serum levels in 56 metastatic melanoma patients, although just 27 patients received ICI. Here, a negative correlation was also seen between high ECP serum levels and the length of OS [51]. Thus, our analyses in a cohort with only melanoma patients receiving ICI provide first evidence that the baseline ECP serum level may be suitable as a prognostic biomarker and that especially a decrease during ICI therapy is prognostically favourable (Figure 6b). Further multicentre studies in metastatic melanoma patients treated with ICI are needed to confirm this finding and, more importantly, to elucidate the underlying mechanisms of the ambivalent ECP effect in vivo. In addition, future studies must be designed to correlate ECP levels in the TME as well as in serum. Only then conclusions can be drawn concerning whether, for example, a high baseline serum ECP level is associated with a lower proportion of infiltrated ECP+ eosinophils in melanoma tissue as these are mobilized into the blood, or instead is linked to a high ECP tissue eosinophilia. With such analyses, it would be possible to explain the different prognostic impact we observed of a high ECP infiltrate in the metastases of the first cohort and elevated baseline ECP level of the second cohort.

Furthermore, our analyses show for the first time an association between increased serum EPX levels and prolonged PFS and OS (Appendix A). EPX is considered a very specific eosinophil marker, so it would be particularly desirable to conduct comparative studies demonstrating its prognostic significance [52,53,54].

In summary, our results suggest a correlation of eosinophil and T-cell tumor infiltration. ECP and CD8 co-staining may provide indications for individual prognosis assessment. In addition, high levels of AEC, ECP and EPX in the blood of metastatic melanoma patients showed prognostic value, suggesting their future potential as serological biomarkers.

According to our data, activated eosinophils are part of the tumor-associated inflammatory microenvironment, and both tumor-infiltrating eosinophils and eosinophil blood counts have prognostic significance. In the future, the mechanisms underlying the positive influence of eosinophils on the prognosis of melanoma patients need to be clarified more precisely. There are first studies pointing to the therapeutic potential of eosinophils to affect tumor progression [55,56]. In syngeneic mouse models of hepatocellular carcinoma and breast cancer, additional treatment with an antidiabetic drug, the dipeptidyl peptidase 4 (DPP4) inhibitor sitagliptin, was shown to promote intratumoral recruitment of eosinophils. Interestingly, this phenomenon was associated with accelerated tumor rejection during ICI treatment [56]. To address the current clinical need for new biomarkers in melanoma and to improve the prognosis of metastatic melanoma patients under ICI, eosinophils should be further investigated as a promising cell population.

## 5. Conclusions

Our data show an association between the extent of eosinophil and T-cell infiltration in melanoma. In particular, a high number of ECP expressing eosinophils and CD8 expressing T-cells in primary melanomas had a favourable impact on prognosis. This suggests that ECP and CD8 as future tissue markers may provide a better prognostic assessment after resection of the primary tumor in melanoma patients based on individual expression patterns. Furthermore, our results reveal the association between elevated AEC, ECP and EPX levels in the peripheral blood and delayed relapse in advanced-stage melanoma patients. For routine clinical practice, measuring blood concentrations of eosinophils, ECP and EPX seem to be relevant prognostic markers.

Overall, eosinophils and their activity markers thus have additional prognostic importance in metastatic melanoma. A deeper understanding of the interaction between eosinophils and T-cells provides the basis for new therapeutic strategies in melanoma patients.

## Figures and Tables

**Figure 1 cancers-14-05676-f001:**
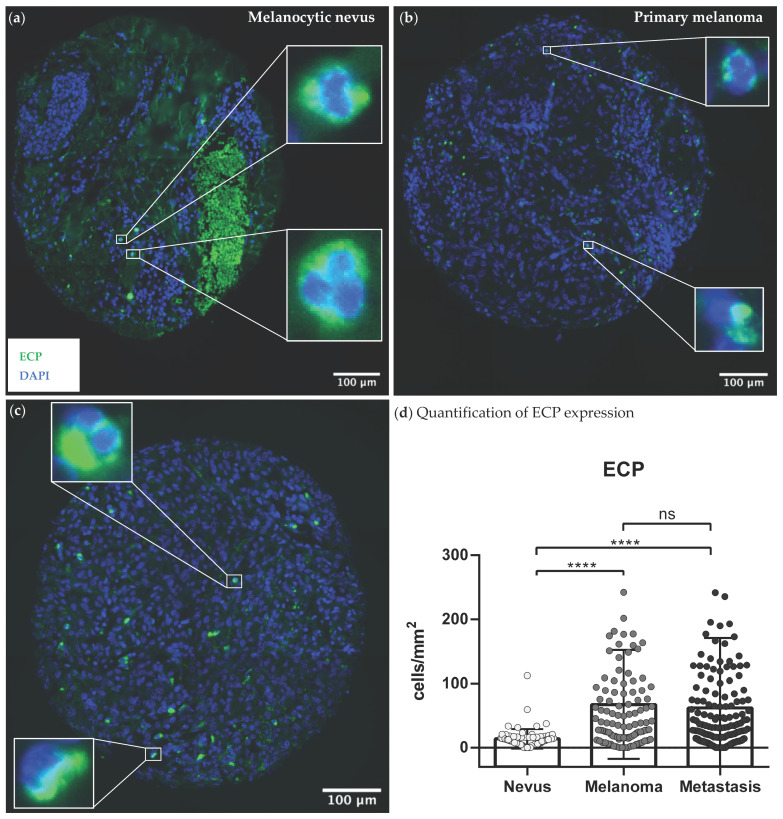
Representative expression patterns of ECP+ eosinophils in human tumor tissue sections. Paraffin-embedded tissue samples of 73 melanocytic nevi, 105 primaries and 151 metastases from 118 melanoma patients were stained with immunofluorescent anti-ECP antibodies (marked green). Nuclei were stained in blue with DAPI. (**a**) Exemplary image of a melanocytic nevi stained with anti-ECP antibodies; (**b**) Exemplary image of a primary melanoma stained with anti-ECP antibodies; (**c**) Exemplary image of an associated metastasis of melanoma stained with anti-ECP antibodies; (**d**) Comparison of ECP expression between nevi, primaries and metastases of melanoma patients. Abbreviations: ECP: eosinophil cationic protein; DAPI = 4′,6-Diamidin-2-phenylindol; ns: not significant; ****: *p* < 0.0001.

**Figure 2 cancers-14-05676-f002:**
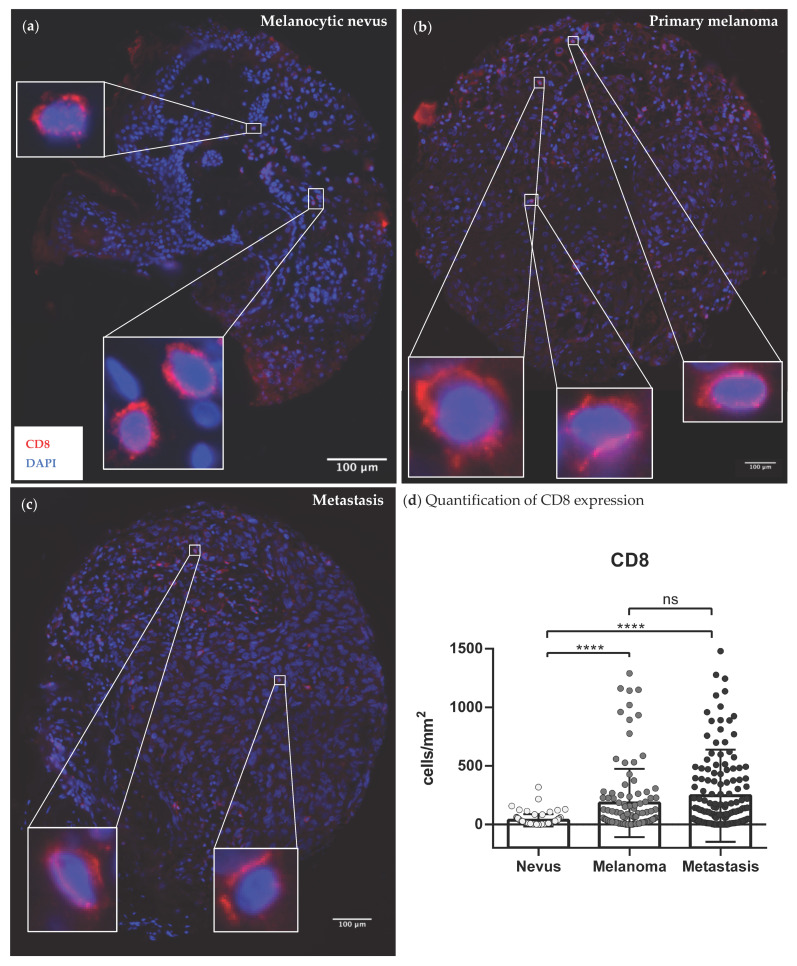
Representative expression patterns of CD8+ effector T-cells in human tumor tissue sections. Paraffin-embedded tissue samples of 77 melanocytic nevi, 108 primaries and 177 metastases from 118 melanoma patients were stained with immunofluorescent anti-CD8 antibodies (marked red). Nuclei were stained in blue with DAPI. (**a**) Exemplary image of a melanocytic nevi stained with anti-CD8 antibodies; (**b**) Exemplary image of a primary melanoma stained with anti-CD8 antibodies; (**c**) Exemplary image of an associated metastasis of melanoma stained with anti-CD8 antibodies; (**d**) Comparison of CD8 expression between nevi, primaries and metastases of melanoma patients. Abbreviations: CD: cluster of differentiation; DAPI = 4′,6-Diamidin-2-phenylindol; ns: not significant; ****: *p* < 0.0001.

**Figure 3 cancers-14-05676-f003:**
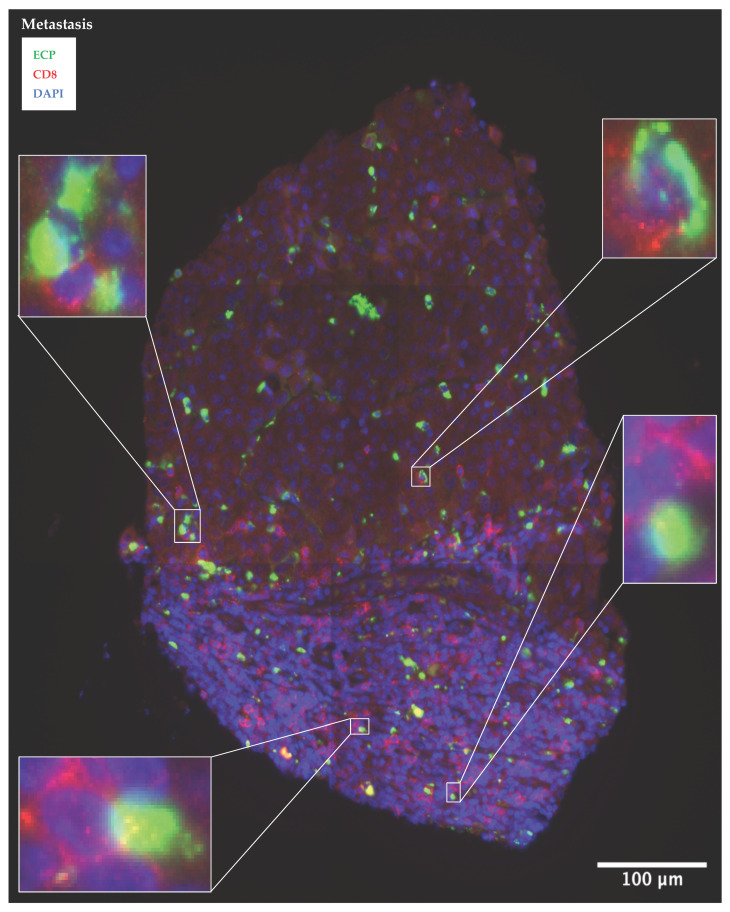
Melanoma metastasis with a high contiguous infiltration of ECP expressing eosinophils and CD8 expressing effector T-cells. Paraffin-embedded tissue samples of a metastasis are co-stained with immunofluorescent anti-ECP antibodies (marked green) and anti-CD8 antibodies (marked red). Nuclei were stained in blue with DAPI. Abbreviations: ECP: eosinophil cationic protein; CD: cluster of differentiation; DAPI = 4′,6-Diamidin-2-phenylindol.

**Figure 4 cancers-14-05676-f004:**
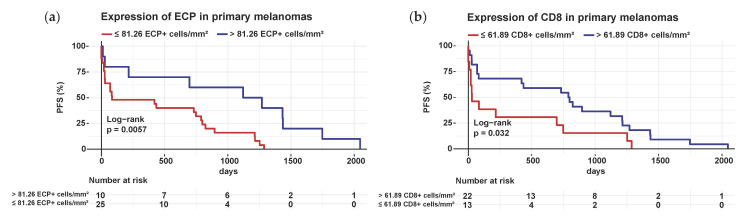
High numbers of ECP expressing eosinophils and CD8 expressing effector T-cells are linked with prolonged PFS in primary melanoma. Kaplan–Meier survival curves for PFS of melanoma patients were stratified by the level (high vs. low infiltration) of ECP+ eosinophils and CD8+ T-cells in the 108 primary melanoma samples. *p*-values were calculated by the two-sided log-rank test. (**a**) Survival analysis of melanoma patients with increased (>81.26 cells/mm^2^) ECP expressing cells in the primaries of melanoma versus those with low ECP expression; (**b**) Survival analysis for PFS of melanoma patients with increased (>61.89 cells/mm^2^) CD8 expressing cells in the primaries of melanoma versus those with low CD8 expression. Abbreviations: PFS: progression-free survival; ECP: eosinophil cationic protein; CD: cluster of differentiation; *p*: *p*-value.

**Figure 5 cancers-14-05676-f005:**
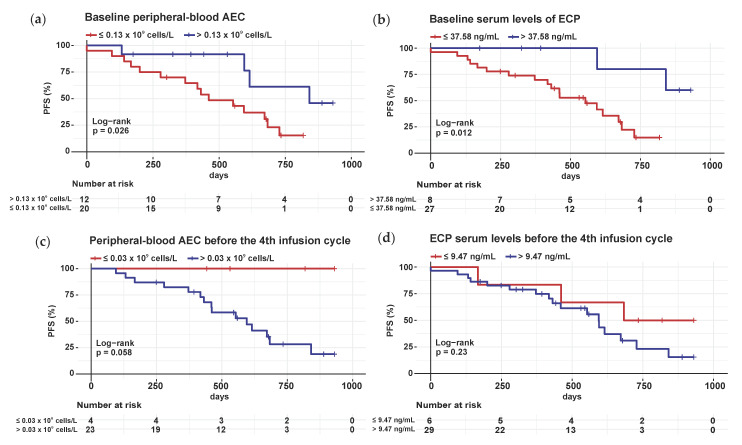
Elevated baseline peripheral blood AEC and ECP serum levels are associated with extended PFS in stage III and IV melanoma patients treated with ICI. Blood was taken before the start of ICI (baseline) and shortly before the respective infusion cycle of ICI (1st cycle, C1; 2nd cycle C2; 3rd cycle C3, 4th cycle C4; etc.). For our concentration measurements, we used serum at baseline and C4; Kaplan–Meier survival curves for PFS of advanced staged melanoma patients were stratified by the amount (high vs. low) of peripheral-blood AEC and ECP serum levels. *p*-values were calculated by the two-sided log-rank test; (**a**) Survival analyses for PFS of advanced melanoma patients with elevated (>0.13 × 10^9^ cells/L) baseline AEC versus those with low peripheral blood AEC; (**b**) Survival analyses for PFS of advanced melanoma patients with elevated (>37.58 ng/mL) baseline ECP serum levels versus those with low ECP levels; (**c**) Survival analyses for PFS of advanced melanoma patients with elevated (>0.03 × 10^9^ cells/L) C4 AEC versus those with low peripheral blood AEC; (**d**) Survival analyses for PFS of advanced melanoma patients with elevated (>9.47 ng/mL) C4 ECP serum levels versus those with low ECP levels. Abbreviations: AEC: absolute eosinophil count; ECP: eosinophil cationic protein; PFS: progression-free survival; ICI: immune checkpoint inhibition; C4: 4th infusion cycle; *p*: *p*-value.

**Figure 6 cancers-14-05676-f006:**
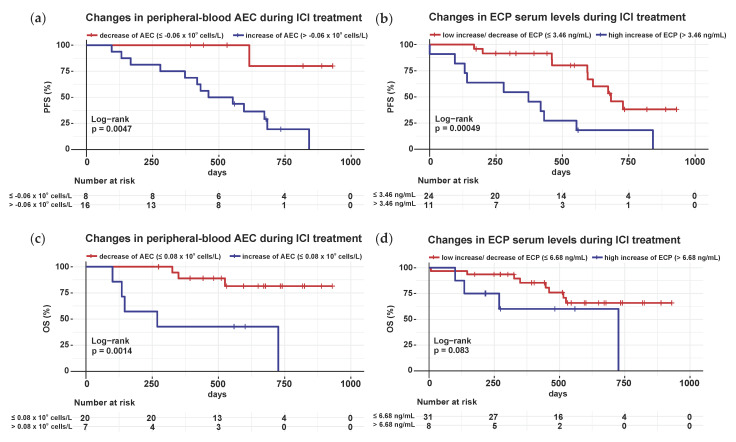
Decreasing peripheral-blood AEC, as well as constant to decreasing ECP serum levels between baseline and the fourth infusion cycle of ICI correlate with extended PFS and OS in met-astatic melanoma. For the kinetic analyses, the level of AEC and ECP before the fourth infusion cycle of ICI (C4) was subtracted from the corresponding level before ICI was initiated (baseline). Kaplan–Meier survival curves for PFS and OS of patients with advanced melanoma were stratified by the extent of the difference between baseline and C4 values of AEC and ECP in blood. *p*-values were calculated by the two-sided log-rank test; (**a**) Survival analyses for PFS of stage III and IV melanoma patients with decreasing (cut-off −0.06 × 109 cells/L) AEC versus those with increasing AEC in peripheral blood between baseline and C4; (**b**) Survival analyses for PFS of stage III and IV melanoma patients with constant to decreasing (cut-off 3.46 ng/mL) ECP serum levels compared to patients with increasing ECP levels between baseline and C4; (**c**) Survival analyses for OS of stage III and IV melanoma patients with decreasing (cut-off 0.08 × 109 cells/L) AEC versus those with in-creasing AEC in peripheral blood between baseline and C4; (**d**) Survival analyses for OS of stage III and IV melanoma patients with low increasing to decreasing (cut-off 6.68 ng/mL) ECP serum levels compared to patients with high increasing ECP levels between baseline and C4. Abbreviations: AEC: absolute eosinophil count; ECP: eosinophil cationic protein; ICI: immune checkpoint inhibition; C4: 4th infusion cycle; PFS: progression-free survival; OS: overall survival; BE: baseline; *p*: *p*-value.

**Table 1 cancers-14-05676-t001:** Detailed overview of the TMA characteristics.

TMA Characteristics	*n*	%
**Localisation of the nevus**	**77**	**21.2**
Trunk	62	80.5
Extremities	15	19.5
**Localisation of the primary melanoma**	**108**	**29.8**
Trunk	78	72.2
Extremities	29	26.9
Unknown	1	0.9
**Localisation of the metastases**	**177**	**48.9**
Locoregional metastases	157	88.7
Subcutaneous/Cutaneous	75	47.8
Lymph nodes	82	52.2
Distant metastases	17	9.6
Unknown	3	1.7
**Histological subtype of the primary melanoma**	**108**	**29.8**
SSM	25	23.1
NMM	15	13.9
LMM	6	5.6
ALM	8	7.4
Other	23	21.3
Unknown	31	28.7
**Breslow thickness of melanoma**	**108**	**29.8**
≤1 mm	13	12.0
1–2 mm	18	16.7
2–4 mm	40	37.0
>4 mm	10	9.3
Unknown	27	25.0
**Ulceration**	**108**	**29.8**
Yes	34	31.5
No	43	39.8
Unknown	31	28.7

Abbreviations: TMA: tissue microarray; *n*: number of samples; SSM: superficial spreading melanoma; NMM: nodular melanoma; LMM: lentigo maligna melanoma; ALM: acro-lentiginous melanoma.

**Table 2 cancers-14-05676-t002:** Line-up of Kendall-Tau correlations between eosinophil markers (Siglec-8, ECP, EPX) and T-cell marker CD8.

Correlation Between	Kendall’s Rank Correlation Coefficient (R)	*p*
**Siglec-8 and CD8**	0.42	<0.0001
**ECP and CD8**	0.34	<0.0001
**EPX and CD8**	0.41	<0.0001

Abbreviations: Siglec-8: sialic acid-binding Ig-like lectin 8; CD: cluster of differentiation; ECP: eosinophil cationic protein; EPX: eosinophil peroxidase; *p*: *p*-value.

**Table 3 cancers-14-05676-t003:** Patient description of the 45 included advanced stage melanoma patients for the blood analyses.

Patient Characteristics	*n*	%
**Age (years)**		
Median (range)	60.9 (20–82)	
**Gender**		
Male	32	71.1
Female	13	2.9
**AJCC (eighth edition)**		
III	4	8.9
IV	41	91.1
**Lines of treatment**		
First line	29	64.4
Second line	14	31.1
IFN therapy	3	21.4
Chemotherapy	2	14.3
*BRAF*i or *BRAF*/MEKi	7	50
Pembrolizumab	2	14.3
Third line	2	44.4
**Administered immune checkpoint inhibitors**		
Ipilimumab + Nivolumab	28	62.2
Nivolumab	6	13.3
Pembrolizumab	11	24.5
** *BRAF* ** ***V600* mutation**		
Yes	19	42.2
No	23	51.1
Unknown	3	6.7
**Immune-related adverse events**		
Grade I or II	16	35.6
Grade III or IV	10	22.2
**Clinical response (according to RECIST 1.1)**		
CR	13	28.9
PR	16	35.6
SD	5	11.1
PD	11	24.4

Abbreviations: *n*: number of patients; AJCC, American Joint Committee on Cancer (8th edition); IFN: Interferon; *BRAF*: B Rapidly accelerated fibrosarcoma; MEK: mitogen-activated protein kinase; RECIST: Response Evaluation Criteria In Solid Tumors; CR: complete response; PR: partial response; SD: stable disease; PD: progressive disease.

## Data Availability

For original data please contact ch.gebhardt@uke.de.

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
