# Peer review of "Activated Eosinophils Predict Longer Progression-Free Survival under Immune Checkpoint Inhibition in Melanoma"

_cancers, 2022, doi:10.3390/cancers14225676_

Round 1
Reviewer 1 Report (Previous Reviewer 2)
I very much appreciate that the authors responded to our questions clearly and made improvements in the quality overall. I don't have any more questions.
Reviewer 2 Report (Previous Reviewer 3)
Authors improved the manuscript very well. No more requests from my side.
Reviewer 3 Report (New Reviewer)
I revised the document and I consider that the authors successfully addressed all the changes suggested during the first round of revisions.
This manuscript is a resubmission of an earlier submission. The following is a list of the peer review reports and author responses from that submission.
Round 1
Reviewer 1 Report
Ammann at all submitted an article titled “Activated Eosinophils Predict Longer Progression-Free Survival under Immune Checkpoint Inhibition in Melanoma”. In the manuscript, they describe two basic studies. In the first study, they correlated the presence of eosinophils with CD8+ cells in nevi, primary and metastatic melanomas using tissue microarrays. While in the other included study, they examined the correlation of serum eosinophil numbers and eosinophil activation markers with melanoma tumor progression in patients undergoing ICI therapy for unresectable stage III or IV melanomas. The manuscript is generally well written and is worthy of consideration for publication. However, some significant limitations and shortcomings are essential to address.
Major points:
In the first part of the study, higher expression of Siglec-8, ECP, EPX, and CD8 in metastases were associated with accelerated progression. This needs to be also discussed in light of the second part of the study. In the second part of the study, in stage III and IV melanomas, the baseline serum features suggestive of eosinophil activation were associated with favorable PFS.
The manuscript contains several overstatements:
Line 541: „Our data indicate that activated eosinophils may support the anti-melanoma T-cell response and thus have a crucial impact on ICI treatment efficacy.” The current study is descriptive and correlative. It does not provide any mechanistic evidence for the role of eosinophils. All instances where the correlative data is overinterpreted need to be corrected.
Eosinophil activation and even eosinophil numbers can be markedly affected by medications. irAE are common during ICI therapy. Steroids and antihistamines may modify the studied serum markers. Therefore, it would be essential to include information on patients who received treatment that may change eosinophil numbers or activity.
Minor points
Line 39: “Here, increased tumor-infiltrating eosinophils and T-cells positively correlated in melanoma associating with tumor regression.” Please correct. The word "here" makes the sentence read as if it referred to the previous sentence and the ICI treatment. Moreover, also correct „regression”. The study did not address melanoma „regression”.
Line 41: „We hypothesize that activated eosinophils facilitate local T-cell infiltration and thereby actively counteract melanoma progression”. This is entirely hypothetical. The study provides no mechanistic evidence for this. Please use a different sentence as the last sentence of the abstract.
Line 445: „The expression of eosinophil surface marker protein Siglec-8 correlated positively with CD8 of the effector T-cells, suggesting a mutual promotion of tumor infiltration (Table 2, Figure S3).” This is only a correlation. The prior literature may suggest that there may be a mutual promotion of tumor infiltration, but the presented data do not provide evidence for the mechanism.
Line 475 „The antitumoral effect of ECP expressing eosinophils in melanoma primaries observed in this study could be explained in two ways”. Please correct. In the published study, there were no antitumoral effects observed, only correlations.
In the tables: instead of „not classifiable” write „unknown”.
Line 492: „Elevated baseline peripheral-blood AEC were found to prolong PFS in 45 unresected stage III and IV melanoma patients”. Please correct to „was found to be associated with...” there is no evidence that AEC actively prolonged PFS.
Line 527: „... receiving ICI give first evidence that...” correct „give”
Line 541: „Our data indicate that activated eosinophils may support the anti-melanoma T-cell response and thus have a crucial impact on ICI treatment efficacy.” However, line 420 states: „Responders and non-responders do not differ in peripheral blood AEC, as well as ECP and EPX serum level”. The data does not seem to support the statement in line 541.
The whole paragraph starting with line 541 and ending with line 558 should be greatly abbreviated. Most of the contents of the section are speculative and should be presented in a shorter format indicating the speculative nature.
Please rewrite lines 559-561 and focus more on the original content of the current manuscript and prognostic markers and the prognostic significance of eosinophils.
Please rewrite the conclusion based on the above comments.
Some figures have illegible (symbol) legends. Likely this is a pdf correction error. Please correct this.
Reviewer 2 Report
In this manuscript, Ammann et.al. performed the immunofluorescence of eosinophil and T cell markers in 285 primary or metastatic 35 tumor tissue specimens from 118 patients post-ICI treatment. Increased tumor-infiltrating eosinophils and T-cells were observed and positively correlated in melanoma associated with tumor regression. Overall, this is certainly a novel and important finding highlighting the crosstalk among tumor-immune pathways with translational applicability.
I have the following comments:
1. Some conclusions are over-interpreted. The authors only observed a correlation between activated eosinophils and CD8+ T-cells in melanoma. None of the cell subtypes, function, or signaling pathway of the two cell types have been examined in this manuscript, but the terms like “eosinophils could aid T-cells mediated immune response against tumor cells” occurs multiple times in the summary and conclusions. If the authors strongly hypothesize and conclude it, the evidence of how they interact in your patient setting should be proved with more data, eg. IFN-r, chemokines.
2. Not all the cited references are relevant to the research and more relevant references could be added. Eosinophil-lymphocyte interactions in the tumor microenvironment and cancer immunotherapy are not novel but not cited, eosinophils can interact with T cells in various ways. Eg.activation of eosinophils with IFN-γ can induce the secretion of multiple chemokines, including CXCL9, CXCL10, and CCL5, which support the recruitment and cytotoxic activities of CD8+ T cells. (Grisaru-Tal etal, 2022 nature immunology)
3. The staining images could be smaller, and the Graphical Abstract could be simpler. Some survival analyses are garbled.
Reviewer 3 Report
General: Authors reported the correlation of activated eosinophils and T cells in melanoma tissue and also discovered the prognostic importance of eosinophils and eosinophils-related activation markers. The manuscript focused on the interesting prognostic role of eosinophils and eosinophils activation markers in association of ICI in melanoma patients considering the role of eosinophils in the ICI-related tumor microenvironment has not been fully studied so far. This study contributes to growing evidence of eosinophils’ role in the ICI treatment.
Main comments:
1) Can authors show the occurrence of irAE in their study? It is because the ICI use can induce eosinophil-related inflammation.
2) Were the counts of eosinophils or activation biomarkers associated with the primary use of other therapies than ICI such as IFNg, chemotherapy, BRAFi/MEKi etc shown in Table 3?
3) Were the counts of eosinophils or activation biomarkers associated with the BRAF V600 mutation status?
Minor comments:
1) Some of the images (Fig4b and Fig5b) looked like garbled characters. Please double-check the figure characters in the other operation system computers.
2) Page19, Line 554 “DDP4” must be “DPP4”.